# Predictive Value of Ultrasound Characteristics for Disease-Free Survival in Breast Cancer

**DOI:** 10.3390/diagnostics12071587

**Published:** 2022-06-29

**Authors:** Qiang Guo, Zhiwu Dong, Lixin Jiang, Lei Zhang, Ziyao Li, Dongmo Wang

**Affiliations:** 1Department of Ultrasound Medicine, Jinshan Branch of Shanghai Sixth People’s Hospital, Shanghai Jiaotong University, Shanghai 201599, China; 2Department of Laboratory Medicine, Jinshan Branch of Shanghai Sixth People’s Hospital, Shanghai Jiaotong University, Shanghai 201599, China; dongzw312@163.com; 3Department of Ultrasound Medicine, Renji Hospital, Shanghai Jiaotong University, Shanghai 201599, China; jianglixin74@gmail.com; 4Department of Ultrasound Medicine, The Second Affiliated Hospital of Harbin Medical University, Harbin 150086, China; zhanglei6@hrbmu.edu.cn (L.Z.); li15846388064@163.com (Z.L.); dongmowang@126.com (D.W.)

**Keywords:** breast cancer, disease-free survival, ultrasonography

## Abstract

The main objective of this study was to determine the predictive value of US characteristics for disease-free survival (DFS) in BC patients. We retrospectively analyzed the ultrasonic images and clinical data of BC patients who had previously undergone breast surgery at least 10 years before study enrollment and divided them into a case group and a control group according to the cutoff value of 120 months for DFS. Correlation analysis was performed to identify US characteristics as independent predictors for DFS by multivariable logistic regression and Kaplan–Meier survival analysis. A total of 374 patients were collected, including 174 patients in the case group with short-DFS and 200 patients in the control group with long-DFS. Three US characteristics (size on US, mass shape, mass growth orientation) and two clinical factors (axillary lymph node (ALN), molecular subtypes) were identified as independent predictors for DFS (*p* < 0.05). The ROC curve showed good performance of the multivariate linear regression model with the area under the curve being 0.777. The US characteristics of large size, irregular shape, and nonparallel orientation were significantly associated with short-DFS, which is a promising supplementary for clinicians to optimize clinical decisions and improve prognosis in BC patients.

## 1. Introduction

Breast cancer (BC) is the most prevalent neoplastic disease among women worldwide [1,2]. An estimate of 287,850 new BC cases and 43,250 deaths cases will occur in the United States in 2022 according to a report on cancer statistics [3]. BC still remained a high rate of recurrence and metastasis even if it is diagnosed and treated early, which leads to short disease-free survival (short-DFS), low quality of life, and high mortality. More targeted cancer control interventions and investment in improved early detection and treatment would facilitate the reduction in cancer mortality [3]. Therefore, it is urgent to establish a convenient and precise prognostic method for short-DFS to improve individualized management for BC patients.

To date, many predictive methods have been used in individual prediction for survival estimation in BC patients, such as the traditional tumor node metastasis (TNM) staging, Magee Equations (MEs) [4,5], and Nottingham Prognostic Index (NPI) [6,7], which were based on pathology and immunohistochemistry results. Furthermore, some genomic methods, including gene expression profiles [8], Adjuvant! Online [9] and the Mamma Print 70-gene expression assay [10], have also been reported to improve the predictive performance for BC survival outcomes. However, those models can hardly avoid the limitations of declined performance, high price, and complex operation, which desperately need some new methods to improve predictive performance for BC prognosis.

Ultrasound (US) is a safe, inexpensive, and widely available modality in breast screening. US image vividly reflects the mass morphological features that provide rich information on growth status, which are not only related to clinical pathology, immunohistochemistry, and molecular subtypes but also associated with invasive ability in BC [11,12]. The significant relationship between US characteristics and BC prognosis has also been evaluated in some papers recently [13,14]. However, despite there being multiple US features that can be used as crucial predictors for the prognosis of BC, it still merits further investigation. Therefore, in this study, we evaluated the US features strongly related to DFS by analysis for ten-year follow-up data in BC patients, which has great promise to further optimize clinical decisions and improve prognosis.

## 2. Materials and Methods

### 2.1. Ethical Approval

The whole research process has been designed to conform to the ethical standard of the 1964 Helsinki declaration. The Ethic Committee of the Second Affiliated Hospital of Harbin Medical University provided an ethical approval (KY2017-133) and waived the informed consent requirement for the retrospective study design.

### 2.2. Patients

This study enrolled 374 BC patients who were treated at the Second Affiliated Hospital of Harbin Medical University from January 2009 to January 2012. According to the cutoff value of 120 months for DFS, patients were divided into a case group with short-DFS and a control group with long-DFS. The inclusion criteria were formulated for the study as follows: (1) a single and unilateral breast cancer based on postoperative pathological findings; (2) BC patients had previously undergone breast surgery for at least ten years before study enrollment; (3) the complete results of pathological diagnosis, immunohistochemistry, and US examination can be obtained from the work station; and (4) having patients’ telephone numbers or e-mail addresses and the follow-up data. We excluded patients who had received any treatments before the operation or suffered from multiple organ metastases. The recurrence or metastasis of cancer-related disease was identified by follow-up results. The DFS was defined as the time interval from post-operation to BC-related relapse or metastasis.

### 2.3. Data and Image Analysis

Preoperative US examinations of B-mode and CDFI were performed using a HITACHI Vision 900 system (Hitachi Medical System, Tokyo, Japan) equipped with a linear-array transducer of 5- to 12-MHz. US variables were collected and analyzed by reviewing each image in work station by two sonographers with more than 5 years of experience in breast US according to the double-blind way. The BI-RADS lexicon [15] and Adler’s grading method of CDFI [16] were used to describe the US characteristics of each breast mass. Nine US characteristics were assessed as follows: lesion size on US (<20 mm or ≥20 mm), shape (oval/round or irregular), margin (circumscribed or not circumscribed), orientation (parallel or nonparallel), echo pattern (hypoecho or others), shadowing features (yes or no), hypoecho surround (yes or no), microcalcification (yes or no), CDFI grade (no flow/minimal or moderate/marked). 

### 2.4. Clinicopathology and Laboratory Examinations

The routine methods of hematoxylin-eosin (HE) stain, formalin-fixed, and paraffin-embedded material were used to perform the histological diagnosis from surgical specimens for tumor histological grades, type of pathology, and the status of the axillary lymph node. Immunohistochemistry analyses were performed to confirm ER, PR, HER2, and Ki-67. Four molecular subtypes of luminal type A, luminal type B, HER2 amplified type and triple-negative type were defined based on the immunohistochemical results.

### 2.5. Statistical Analysis

SPSS version 18.0 statistical software package (IBM Corporation, Armonk, NY) was used for all variables in this study. Chi-square or Fisher’s exact tests were used to identify the potential risk factors of DFS in the categorical variables between the case group and control group. Variables with statistically significant association (*p* < 0.05) in the Chi-square or Fisher’s exact tests were further evaluated by the multivariable logistic regression analyses to establish a regression model for predicting DFS. An area under the curve (AUC) of the receiver operating characteristic (ROC) curve was calculated to evaluate the performance of the prediction model. Kaplan–Meier survival analysis was performed to determine the association of US features and DFS by survivorship curves. All statistical tests were two-sided, and *p* value < 0.05 was considered statistically significant.

## 3. Results

This section is divided into subheadings. It provides a concise and precise description of the experimental results, their interpretation, as well as the experimental conclusions that can be drawn.

### 3.1. Clinical Characteristics

The clinical data of 374 patients (age 33–72 years; mean 56.0 years) were collected and analyzed respectively, including a case group of 174 patients with short-DFS and a control group of 200 patients with long-DFS. There were no significant differences in 11 variables including age, BMI, pathological type, mass margin, shadowing, hypoecho surround, calcifications, echo pattern, ER, Her2, and Ki-67 between the two groups (*p* > 0.05). In terms of molecular subtypes of luminal B or HER2-enriched compared with luminal A, no significant difference was observed between the two groups (*p* > 0.05), while a statistical difference was obtained in TN subtype compared with luminal A between the two groups (*p* < 0.001). There were significant differences in seven variable parameters including the status of ALN, nuclear grade, size on US, mass shape, mass orientation, CDFI, PR between two groups (*p* < 0.05). The correlation analysis is summarized in Table 1.

### 3.2. Multiple Logistic Regression Analysis

For rigorous variable selection, the risk factors with a *p*-value < 0.05 in the simple logistic regression analysis were considered for a multiple logistic regression model. The parameters of size on US, status of ALN, mass shape, mass orientation, and molecular subtypes were identified as independent predictors for DFS and the results are summarized in Figure 1. With respect to the tumor size on US, patients with large tumor size were more likely to have short-DFS compared with those with small size (X1, OR: 1.930, 95% CI: 1.209–3.082, *p* = 0.006). Patients with negative ALN were least likely to have short-DFS compared with those with positive ALN (X2, OR: 0.231, 95% CI: 0.142–0.375, *p* < 0.001). Ultrasound characteristic of irregular shape was a higher risk factor than round or oval shape in the patients with short-DFS (X3, OR: 2.052, 95%CI: 1.284–3.280, *p* = 0.003). Moreover, patients with ultrasound features of nonparallel growth orientation were more likely to suffer from short-DFS than patients with parallel growth orientation (X4, OR: 1.573, 95%CI: 1.077–2.297, *p* = 0.019). In the four molecular subtypes, the TN subtype showed a higher risk for short-DFS compared to luminal A (X5, OR: 1.669, 95%CI: 1.364–2.042, *p* < 0.001). According to the constant of -1.773, a multivariate regression model was established as follows: y = −1.773 + 0.657 * X1–1.466 * X2 + 0.719 * X3 + 0.453 * X4 + 0.512 * X5, as shown in Table 2. The ROC was used to evaluate the performance of the regression model and the AUC was 0.777, which is shown in Figure 2.

### 3.3. Survival Analysis for DFS

Kaplan–Meier survival curves were drawn using disease-free survival time and survival status of patients and the differences between disease-free survival rates were compared by log-rank tests. Each of the independent predictive factors, including size on US (<20 mm vs. ≥20 mm), status of ALN (positive vs. negative), mass shape (round, oval vs. irregular), mass orientation (parallel vs. nonparallel), molecular subtypes (luminal A, luminal B, HER2-enriched vs. triple negative), had a significant statistical difference in terms of affecting the DFS (Logrank *p* < 0.001), which were illustrated in Figure 3.

## 4. Discussion

In this study, we drew an interesting conclusion that the tumor US characteristics of size on US, shape, and growth orientation are associated with DFS in BC patients. Furthermore, a prediction model combining US characteristics with clinical predictors of ALN and molecular subtypes was established and showed good predictive performance. Therefore, the application of US features as a predictor for DFS suggests a new idea that morphological characteristics of tumors are related to prognosis, which can be served as an effective complement to clinical features for prognostic prediction in BC.

Our study showed that there was a significant relationship between tumor size and DFS in BC patients, and the patients with large-sized tumors are inclined to have a short DFS. The tumor size in BC patients is an important prognostic factor that has been affirmed by the TNM cancer staging system [17]. Some studies also concluded that tumor size has been thought of as an indispensable prognostic factor in the clinical outcome of BC [18,19,20]. Conventionally, tumor size is measured as a diameter based on palpation, reviewing an image, or measuring a surgical specimen. However, the size on US is relatively accurate and convenient because of its high resolution, multiple aspects, and cine clips. Biologically, tumor size reflects the number of cancer cells with invasiveness and metastasis capability. Moreover, the number of cancer cells increases eight-fold for every doubling of tumor size [21], which would lead to the fact that large tumor may have more chances to invade the adjacent tissue and cause metastasis than a small one, making it difficult to perform complete resection. The large surface of the tumor is a significant influence factor in the metastasis of cells, which enhances the surface contact area between cancer cells and adjacent tissue, and improves the capability of spread to other areas of the body. This was mainly attributed to the larger surface area of the tumor with a large diameter according to the formula of a sphere area (S = 4πr^2^, where r is the radius). Furthermore, our study showed that the patients with tumor diameters of more than 20 mm have a significant short-DFS in BC (a high OR value 1.930). Therefore, it is of great significance to use the tumor size combined with US features as a model for the prediction of DFS in BC.

Irregular shape can further increase the surface area of tumor contact with surrounding normal tissue compared with the same size tumor, which leads to more carcinoma cells dislodging from the surface into metastasis. Therefore, the shape of the tumor is also a critical factor in the evaluation for the prognosis of BC. However, the characteristic of shape was rare to be used as a predictive factor in BC patients because it is difficult to evaluate in surgical specimens. US is suitable to evaluate the morphometric properties of a tumor by cine clips through the masses with a longitudinal axis and a transverse axis [22]. We concluded that the US characteristic of irregular shape was a higher risk factor for the patients with short-DFS compared with a round or oval shape with an OR value of 2.052. Irregular shape is a critical property in BC [23], which represents not only the nonuniform growth speed of mass edge but also the highly invasive ability of cells [24]. Furthermore, surface area is the exposure range of neoplastic tissue coming in contact with normal tissue, which is much greater and increases rapidly in tumors with irregular shapes and convoluted surfaces [21]. Some research has shown that irregular shape is related to poor prognosis in invasive BC, similar to the result of our study [13,25]. Therefore, the shape features in US would be an amazing predictive factor for the prognosis of BC.

Interestingly, the feature of nonparallel growth orientation on US was found to be an independent predictor of the prognosis in BC patients in our study, with an OR value of 1.573 for short-FDS. Nonparallel orientation is explained as the mass growth perpendicular to the skin surface and shape taller than wide according to the ACR Reporting system [26]. The characteristic of vertical growth represents a sign that carcinoma cells can easily destroy normal breast tissue growing orienting skin and have highly aggressive characteristics and a grave prognosis [27]. Wang et al. found that nonparallel growth orientation was similar to molecular subtypes on the prognostic values for BC and can be served as a prognostic biomarker for triple negative BC patients [28]. Our previous study also demonstrated that the vertical growth orientation of a tumor in preoperative US examination was associated with a high recurrence risk of BC [14]. Furthermore, a study found that vertical growth orientation be related to axillary lymph node metastasis in TNBC patients [29]. Therefore, nonparallel growth orientation is a promising predictive factor for prognosis of BC patients, which is easy to be evaluated by preoperative US examination.

Clinicopathologic factors including ALN and molecular subtypes have been commonly proved to be significantly associated with the prognosis of BC [30,31,32], which is similar to our study. In recent years, using US features as predictive factors in BC prognosis has been reported in some research [14,29,33]. However, incorporation of US features and clinical factors was rarely performed in terms of predicting short-DFS in a ten-year follow-up study. Our study showed good predictive performance (AUC = 0.777) of US characteristics for the DFS and put forward a new idea of applying morphometrical features of tumors as prognostic predictive factors in the BC patients.

## 5. Conclusions

In a 10-year follow-up study, we concluded that US characteristics of large size, irregular shape, and nonparallel orientation were significantly associated with short-DFS. The cooperation of US characteristics and clinical factors is a promising supplementary in predicting DFS for clinicians to optimize clinical decisions and improve prognosis in BC patients.

However, some limitations in our research are as follows: firstly, our study is a retrospective and single-centered work with unavoidable bias in the basic characteristics of the study population; secondly, the sample size was relatively small in the short-DFS group, which should be further confirmed by a large population; thirdly, some genomic characteristics and gene markers were not considered in this research; fourthly, the US as a highly operator-dependent technology performed by different sonographers can influence the results of image analysis. Moreover, a combination of multiple image technologies including US, MRI, and contrast enhancement spectral mammography for the same tumor will certainly be of great benefit in further validating our findings and improving the prediction performance. Further work in the area will be needed to conquer these limitations.

## Figures and Tables

**Figure 1 diagnostics-12-01587-f001:**
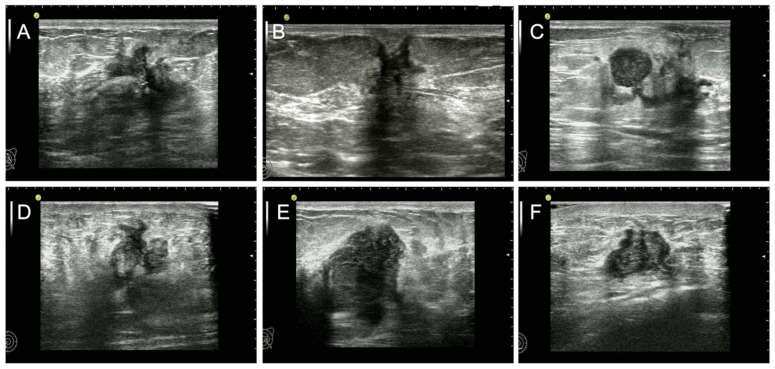
US feature as independent predictor for short-DFS. We displayed the US images of breast mass from preoperative examination in female patients with invasive ductal carcinoma who suffered from surgical treatment and received postoperatively systemic therapy. (**A**) The image of the left breast mass from a 48-year-old patient with a short-DFS of 79 months shows the US features of significantly irregular shape and moderately parallel growth orientation. (**B**) The image of the right breast mass from a 44-year-old patient with a short-DFS of 51 months shows the US features of significantly irregular shape and slightly nonparallel growth orientation. (**C**) The image of the left breast mass from a 55-year-old patient with a long-DFS of more than 120 months shows the US features of round shape and slightly parallel growth orientation. (**D**) The image of the right breast mass from a 59-year-old patient with a short-DFS of 58 months shows the US features of significantly nonparallel growth orientation and moderately irregular shape. (**E**) The image of the right breast mass from a 42-year-old patient with a short-DFS of 35 months shows the US features of significantly nonparallel growth orientation, oval shape, and size of more than 20 mm. (**F**) The image of the right breast mass from a 58-year-old patient with a long-DFS of more than 120 months shows the US features of oval shape, and parallel growth orientation.

**Figure 2 diagnostics-12-01587-f002:**
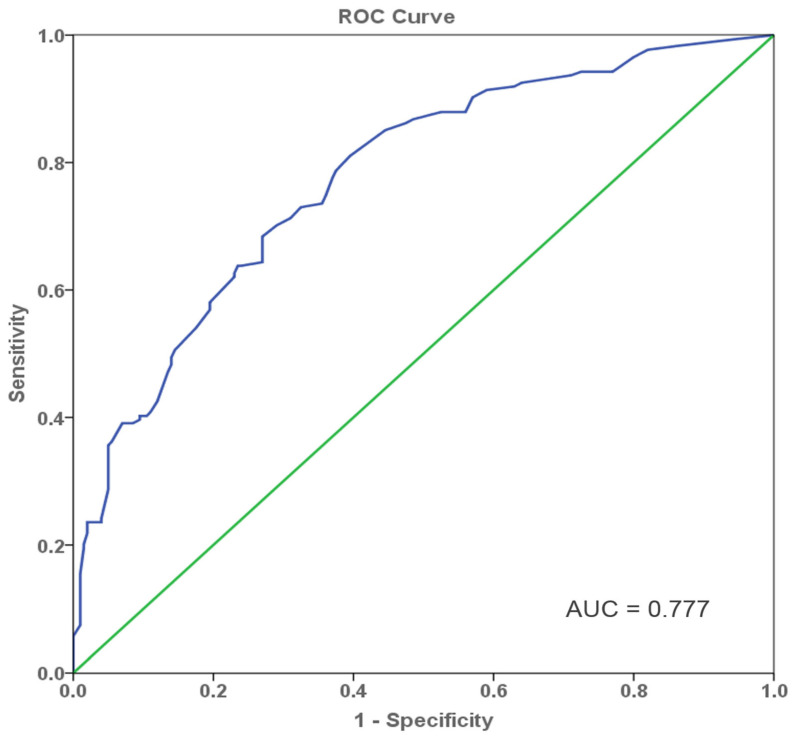
ROC curve for the predictive model. Analysis of the predictive performance of the model for short-DFS in BC patients demonstrated a good discriminative power with an area under the ROC curve of 0.777.

**Figure 3 diagnostics-12-01587-f003:**
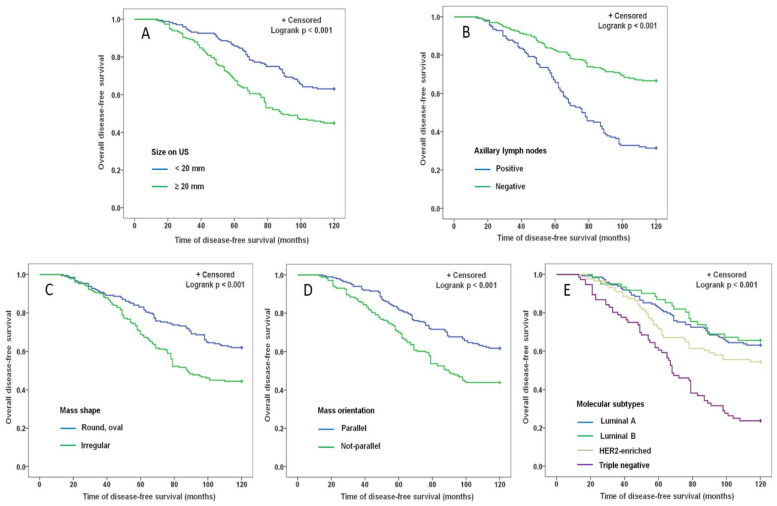
Kaplan–Meier survival analysis curve for each predictive factor in BC. The X-axis represents the following-up time in 120 months and the Y-axis represents DFS rate at different times. Each of the independent predictive factors affecting the DFS had a significant statistical difference in features (Logrank *p* < 0.001), including size on US (**A**), status of ALN (**B**), mass shape (**C**), mass orientation (**D**), molecular subtypes (**E**).

**Table 1 diagnostics-12-01587-t001:** Patient characteristics and univariable analysis of factors associated with disease-free survival.

Variables	Disease-Free Survival		*p* Value	Odds Ratio (95% CI)
	<Ten Years *n* = 174, (%)	≥Ten Years *n* = 200, (%)		
Age (years)			0.405	1.21 (0.81–1.82)
<50	90 (51.7)	113 (56.5)		
≥50	84 (48.3)	87 (43.5)		
BMI (kg/m^2^)			0.123	1.43 (0.93–2.20)
<25	109 (62.6)	141 (70.5)		
≥25	65 (37.4)	59 (29.5)		
Pathological type			0.062	0.67 (0.45–1.01)
IDC	100 (57.5)	95 (47.5)		
Others	74 (42.5)	105 (52.5		
Nuclear grade			0.009	1.77 (1.15–2.71)
Low/Intermediate	101 (58.0)	142 (71.0)		
High	73 (42.0)	58 (29.0)		
Size on US (mm)			0.001	2.09 (1.38–3.17)
<20	65 (37.4)	111 (55.5)		
≥20	109 (62.6)	89 (44.5)		
ALN			< 0.001	0.23 (0.15–0.36)
Positive	96 (55.2)	44 (22.0)		
Negative	78 (44.8)	156 (78.0)		
Mass shape			0.001	2.03 (1.34–3.06)
Round, oval	74 (42.5)	120 (60.0)		
Irregular	100 (57.5)	80 (40.0)		
Mass orientation			0.001	2.06 (1.36–3.12)
Parallel	77 (44.3)	124 (62.0)		
Not-parallel	97 (55.7)	76 (38.0)		
Mass margin			0.334	1.25 (0.82–1.90)
Circumscribed	59 (33.9)	78 (39.0)		
Not-circumscribed	115 (66.1)	122 (61.0)		
Shadowing			0.144	0.73 (0.49–1.11)
Yes	82 (47.1)	79 (39.5)		
No	92 (52.9)	121 (60.5)		
Hypoecho surround			0.524	1.16 (0.768–1.77)
Yes	65 (37.4)	82 (41.0)		
No	109 (62.6)	118 (59.0)		
Calcifications on US			0.078	0.69 (0.46–1.04)
Yes	90 (51.7)	85 (42.5)		
No	84 (48.3)	115 (57.5)		
Echo pattern			0.412	0.82 (0.51–1.30)
Hypoechoic	132 (75.9)	144 (72.0)		
Others	42 (24.1)	56 (28)		
CDFI			0.002	2.15 (1.33–3.48)
No flow, Minimal	33 (19.0)	67 (33.5)		
Moderate, Marked	141 (81.0)	133 (66.5)		
ER			0.178	1.34 (0.89–2.02)
Positive	82 (47.1)	109 (54.5)		
Negative	92 (52.9)	91 (45.5)		
PR			0.014	1.73 (1.14–2.63)
Positive	95 (54.6)	135 (67.5)		
Negative	79 (45.4)	65 (32.5)		
HER2			0.159	0.73 (0.48–1.13)
Positive	67 (38.5)	63 (31.5)		
Negative	107 (61.5)	137 (68.5)		
KI67			0.063	0.65 (0.41–1.02)
Positive	55 (31.6)	46 (23.0)		
Negative	119 (68.4)	154 (77.0)		
Molecular subtypes			<0.001	
Luminal A	55 (31.6)	94 (47.0)		
Luminal B	21 (12.1)	40 (20.0)	0.755 *	1.11 (0.60–2.08)
HER2-enriched	40 (23.0)	48 (24.0)	0.218 **	0.70 (0.41–1.20)
TN	58 (33.3)	18 (9.0)	< 0.001 ***	0.18 (0.10–0.34)

Note: *p* value is derived from the univariable association analyses between each of the variables and groups. *p* * Luminal B VS. Luminal A, *p* ** HER2-enriched VS. Luminal A, *p* *** TN VS. Luminal A. Abbreviations: ALN, axillary lymph node; BMI, body mass index; CDFI, color doppler flow imaging; ER, estrogen receptor; HER2, epidermal growth factor receptor 2; KI67, Ki-67 Protein; PR, progesterone receptor; TN, triple negative.

**Table 2 diagnostics-12-01587-t002:** Multiple logistic regression analysis of the US characteristics with disease-free survival.

Variable	Β	SE	Wals	*p* Value	Odds Ratio (95% CI)
Size on US (mm)					
<20 VS. ≥20	0.657	0.239	7.581	0.006	1.930 (1.209–3.082)
ALN					
Positive VS. Negative	−1.466	0.248	34.919	<0.001	0.231 (0.142–0.375)
Mass shape					
Round, oval VS. Irregular	0.719	0.239	9.017	0.003	2.052 (1.284–3.280)
Mass orientation					
Parallel VS. Not-parallel	0.453	0.193	5.492	0.019	1.573 (1.077–2.297)
Molecular subtypes					
Luminal A VS. TN	0.512	0.103	24.733	<0.001	1.669 (1.364–2.042)
Constant	−1.773	0.789	5.037	0.025	

Note: *p* value is derived from the univariable association analyses between each of the variables. Abbreviations: US, ultrasound; ALN, axillary lymph node; TN, triple negative.

## Data Availability

The data presented in this study are available on request from the corresponding author. The data are not publicly available due to privacy or ethical restrictions.

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
