# Peer review of "Predictive Value of Ultrasound Characteristics for Disease-Free Survival in Breast Cancer"

_diagnostics, 2022, doi:10.3390/diagnostics12071587_

Round 1
Reviewer 1 Report
I am grateful for the opportunity to review manuscript with ID diagnostics-1770121, entitled “Predictive value of ultrasound characteristics for disease-free survival in breast cancer”.
The authors have produced a clearly written manuscript describing their retrospective analysis of US scanned data from a cohort of 374 previously treated breast cancer patients. The methodology is clearly explained, the results are well presented, and the conclusions are supported by the data. The authors show a correlation between several US characteristics and short- vs. long-DFS, concluding that US imaging may serve as a useful tool in establishing prognosis. The authors also admit limitations of the study and suggest further investigation.
This work is interesting and potential useful. This manuscript will be a useful addition to the literature.
I only have minor comments:
· Page 9: 4pir2 would be better if written as 4πr2
· Top of page 10: Misplaced paragraph to be removed.
Reviewer 2 Report
Dear authors,
thank you for your submission on the interesting research topic about the prognostic methods to improve breast cancer management.
I found your findings very interesting. I also appreciated the methodology you have followed to detect the tumors’ features by reviewing US images (….by reviewing each image in work station by two sonographers with more than 5 years of ex perience in breast US according to double-blind way). US is a very operator-dependent diagnostic technique. I only suggest to highlight this limit in the discussion, because it could influence the use of your results in the clinical practice, where US images are catch by sonographers with different skills and experience.
You could also envisage future research perspectives to validate your findings (e.g. the use of ABVS images, the comparison of the same tumor features drawn by other type of images as those from MRI or CESM, and so on).
I hope that my suggestions would be helpful.
